# Investigating the Design of an Asynchronous Online Discussion (AOD) in Distance Education: A Cooperative Learning Perspective

Tianxiao Yang [1],* and Zhijuan Niu [2],*

1 College of Education, Texas Tech University, Lubbock, TX 79409, USA
2 School of Education, Syracuse University, Syracuse, NY 13210, USA
* Correspondence: tianxyan@ttu.edu (T.Y.); zhniu@syr.edu (Z.N.)

**Abstract:** An asynchronous online discussion (AOD) is considered a commonly used cooperative learning activity in distance education. However, few studies have explored whether AODs are designed in accordance with the conditions of cooperative learning and whether students are able to achieve higher levels of cognitive learning through interactions in AODs. This case study explored if an AOD was designed to meet cooperative learning conditions and whether students generated interactions and accomplished higher levels of cognitive learning. The results suggested that in an AOD where cooperative learning conditions were rarely met, students barely interacted and only manifested lower levels of cognitive learning. The researchers proposed that an AOD may not achieve the expected cooperative learning outcomes unless it is well-structured with a systematical integration of cooperative learning theory.

**Keywords:** asynchronous online discussion; cooperative learning; distance education





## 1. Introduction

The increase in distance education has led to the proliferation and use of asynchronous online discussion (AOD) [1]. AOD is a computer-mediated communication (CMC) activity that allows students to interact with each other at different times, mostly in a text-based form [2–4]. Many researchers have argued that AOD was born to be a cooperative learning activity in distance education due to its unique property [5]. Cooperative learning requires social interactions among learners [6–8], and AOD provides students with such an environment for interactions without time and location restraints [9,10]. Thus, AOD has been regarded as a cooperative learning activity for flexible interactions.

However, few people have noticed that cooperative learning is not a simple terminology emphasizing the occurrence of social interactions. It is an integral instructional theory with a series of conditions that have to be considered and integrated into the instructional design process [6]. The educational practices of cooperative learning require at least four conditions: (1) students are expected to achieve higher levels of cognitive learning and social interactions as learning outcomes; (2) students cooperate with each other in small groups by investing in equal efforts to aim at a common goal; (3) activities should satisfy at least five social interdependence principles including positive interdependence, individual accountability, promotive interaction, social skills, and group processing; (4) activities should be delivered through structures [6–8,11].

Although cooperative learning theory has been well-developed, there have been few studies observing AODs through a lens of cooperative learning theory, not to mention examining whether the AOD design cases meet the conditions of cooperative learning and if the expected cooperative learning outcomes genuinely occur in AODs. It is still common to call AOD a "cooperative learning activity", but no one has explained if each AOD meets the relevant conditions. To fill these gaps in the current literature, the authors externally

observed and investigated AOD in an online course from a comprehensive perspective of cooperative learning theory. The research questions included the following:

1.  What features make the AOD case design consistent with cooperative learning conditions?
2.  What patterns can be found about the expected cooperative learning outcomes (i.e., interactions and higher levels of learning) in the AOD?

## 2. Theoretical Framework: Cooperative Learning

### 2.1. Expected Learning Outcomes

Whichever instructional theory is applied to the instructional design process, the expected learning outcomes should be clearly defined [12]. The expected learning outcomes of cooperative learning can be detected from its definition of it. Cooperative learning is an instructional theory referring to the use of small groups to maximize students' cognitive learning through social interactions by requiring each group member to accomplish common goals through equivalent cooperative efforts [6,8,13,14]. The description of "maximize students' cognitive learning through social interactions" distinguishes cooperative learning from other instructional theories regarding its expected learning outcomes. Rather than emphasizing the importance of achieving subject-relevant skills (e.g., mathematical, medical, engineering), cooperative learning focuses on improving students' general cognitive competency and social interactions [8,11,15]. Through cooperative learning activities, students should achieve higher levels of cognitive learning while interacting with each other frequently.

### 2.2. Requirements

Apart from the expected learning outcomes, the definition of cooperative learning also reveals three essential requirements. First, students must be organized into small groups. Although there has yet to be an agreement about the most appropriate small group size for a cooperative learning activity, there are some suggested sizes as references. Oxford (1997) [8] posited that the size of a cooperative learning group should be less than seven. Some researchers recommended that an ideal small group was supposed to contain two to five members [2]. Second, students in a group should share common goals. These common goals should be observable units, objects, or artifacts and the final products after AODs [11]. Third, the required cooperative efforts from group members should be equal. The group members are supposed to be assigned similar tasks to achieve common goals [16]. If group members are assigned different tasks with a varied workload, the engagement of the students who perform the substantive share of the work decreases [14,17].

These three essential requirements provide the foundation for cooperative learning. However, simply organizing students in small groups, setting shared common goals, and assigning them equal tasks could not guarantee higher levels of cognitive learning with interactions [11,18]. Some students could still reject interacting with peers, even in a small group, as they pay more attention to competition than cooperation [11]. In addition, low-ability students could give up contributing their efforts to the group as they realize their high-ability peers complete the task independently. Meanwhile, high-ability students could also contribute fewer efforts to the group when they are demotivated by their low-ability peers' low engagement [14]. Thus, group interactions could still be very limited and stay at lower levels of cognitive learning.

Hence, more conditions are required to be considered and integrated into the instructional design of a cooperative learning activity. An instructional design does not simply draft an instructional activity by accumulating requirements. It is a "systematic and reflective process of translating principles of learning and instruction into plans for instructional materials, activities, resources, and evaluation" [12]. Thus, to prompt students to interact with each other to achieve higher levels of cognitive learning, previous researchers have summarized five cooperative learning principles. These principles could be translated into the process of instructional design: (1) positive interdependence, (2) individual accountability, (3) promotive interaction, (4) social skills, and (5) group processing [13].

### 2.3. Advanced Principles

Positive interdependence. Positive interdependence derives from the social interdependence theory: the theoretical pillar of cooperative learning. According to social interdependence theory, there are three types of interdependence among group members: positive interdependence, negative interdependence, and no interdependence. Positive interdependence means that group members must rely on each other's efforts to succeed in cooperation [6,14,19]. Positive interdependence can be maximized when three types of it are present: reward interdependence means interdependence and boundary interdependence. Positive reward interdependence means group members stick to their solidarity when they are assigned the same grade for their performance. Positive means interdependence indicates the situation in which all group members rely on each other's resources and roles to pursue the expected learning outcomes [13]. Finally, positive boundary interdependence refers to students who successfully find their group identity as they know what makes their group distinct from other groups [11].

Individual accountability. Individual accountability means that everyone's performance should be visible to all the group members so that every team member can realize their responsibilities in the group. When an individual realizes his performance is visible to the whole group, the individual becomes more self-regulated in fulfilling their duty in the group work [20]. Students' accountability was positively correlated with their positive interdependence [14], thus impacting students' learning outcomes in group work. Individual accountability can be confirmed when a team member's performance is assessed or documented, and the results are sent back to both the individual and the whole group. Reviewing the assessment results, the group members can reflect on whether they performed well in the group work and which groupmates need support in the next round of cooperation [13].

Promotive interaction. As mentioned at the beginning, interaction is one of the expected cooperative learning outcomes. Interaction refers to the information exchange process between learners [8]. Promotive interaction means students are motivated to extend meaningful conversations. Promotive interaction occurs when students are heterogeneously grouped or receive advisement feedback from others [14]. Students who are grouped with peers of different backgrounds or ability levels tend to be motivated to expand the interaction because their diverse perceptions and values are respected [14]. Providing advisement feedback means that the activity or instruction should contain advice about constructing meaningful dialogues using the content from materials and peers [21]. Advisement feedback can help students concentrate on investing in learning during the interaction rather than simply completing the interaction steps [14].

Social skills. Apart from being motivated to create meaningful interactions, students also need social skills to interact effectively. Social skills mainly comprise leadership, decision-making, trust-building, communication, and conflict-management strategies. Being equipped with these social skills helps students extend meaningful interactions in group work [13]. Moreover, the appropriate use of social skills builds up social relationships, strengthening positive interdependence among group members [22]. Many social skills have been available and tested in the previous literature, such as sending reminders to group members (leadership), negotiating until all the group members reached a consensus (decision-making), posing follow-up questions (communication), sharing personal experiences (trust-building), and explaining controversy as a mutual problem (conflict-management) [13].

Group processing. Group processing refers to each group reflecting on their teamwork process to determine what member actions are kept, avoided, or adjusted. As a metacognitive strategy, group processing can directly benefit from the cognitive learning of group members [23]. Group processing can also produce the compensation effect, which suggests that group members can work harder to cover the reported shortcomings in group processing [24]. It also can make the group members who rarely contribute to the group work realize the importance of their engagement in cooperation and become more active

in the next rounds of cooperation [18]. Thus, when group processing is open to all the group members, they become more united, and their social relationship is optimized. This cooperative status can last even after the instruction is over [22].

### 2.4. Activity Structures

Looking into the practices of translating cooperative learning requirements and principles into the classroom, researchers found that some required content-free behaviors repeatedly occurred in the classroom. The sequential uses of such behaviors were defined as cooperative learning structures [8,11,25]. "Structure" here refers to a set of sequential steps that are required for an instructional activity. Employing a structure makes an instructional activity more manageable and then easier to be designed for meeting multiple conditions of an instructional theory [8]. Cooperative learning structures can be varied depending on which cooperative learning requirement or principle is emphasized. There are at least six types of structures: the group-building structure (e.g., Roundtable), communication-creating structure (e.g., Paraphrase Passport), mastery structure (e.g., Numbered Heads Together), cognitive development structure (e.g., Think-Pair-Share), division of labor structure (e.g., Jigsaw), and project structure (e.g., Cop-Cop) [8,15,25].

In summary, cooperative learning is a systematic instructional theory with clear expected learning outcomes, essential requirements, solid instructional principles, and practical activity structures. Figure 1 indicates a cooperative learning activity should be evaluated or designed according to these well-developed conditions.

| Expected Outcomes | Essential Requirements | Advanced Principles | Activity Structures |
|---|---|---|---|
| • Higher Levels of Cognitive Learning<br>• Social Interactions | • Small Group<br>• Equivalent Efforts<br>• Common Goals | • Positive Interdependence<br>• Individual Accountability<br>• Promotive Interaction<br>• Social Skills<br>• Group Processing | • Roundtable<br>• Paraphrase Passport<br>• Numbered Heads Together<br>• Think-Pair-Share<br>• Jigsaw<br>• … |
| (Yang, 2023; Johnson & Johnson, 2009) | (Tamimy et al., 2023; Johnson et al., 2014) | (Johnson et al., 2014; Oxford, 1997) | (Kagan, 1989; Johnson et al., 2014) |

**Figure 1.** Cooperative learning theory [6,8,11,13,17,25].

### 2.5. AOD as a Cooperative Learning Activity

In distance education, AOD was often considered a classic cooperative learning activity for multiple reasons. First, a discussion is a learning process to achieve higher levels of learning through interactions so that it is potentially effective in helping students to meet the expected cooperative learning outcomes [1,9]. The use of technology makes the asynchronous discussion more accessible for students [10]. In addition, there was an argument insisting that the discussion board was a natural environment for the growth of students' positive interdependence, individual accountability, and group processing [2,5]. Many studies tacitly agreed that AOD was a one-shot treatment for facilitating students' cooperative learning in online education. However, most of these studies lacked empirical

evidence proving the fulfillment of cooperative learning in the AOD. Few researchers defined AOD as a cooperative learning activity by verifying the required conditions.

*2.6. The AOD Design*

Instructors and researchers attempted to promote the AOD design by developing different instructional interventions that were applied during the AOD process. The common interventions included prompts, facilitations, roles, and scaffolds. A discussion prompt is a specific rationale that initiates an AOD [4]. Many studies have examined the effects of different prompt formats (e.g., question, case) on students' learning [26,27]. Facilitation is a behavior or a set of behaviors motivating students to exchange information (Rovai, 2007) [28]. There are at least three facilitation strategies in designing an AOD: instructor-led facilitation, student-led facilitation, and co-facilitation [28]. Some instructors required students to play different roles in the AOD. Students were assigned to either discipline-specific roles with fictional identities (e.g., HR manager, sales representative) or generic roles with assigned discussion responsibilities (e.g., moderator, synthesizer) [27,29]. Additionally, scaffolds repeatedly occurred in studies on AODs. Founded by Wood et al. (1976) [30] and Vygotsky (1978) [31], scaffolding refers to any instructional support that assists students in completing learning objectives in the zone of proximal development. Various scaffolds were designed and developed for promoting AODs, such as message labels, posting examples, and discussion guidelines [29].

Previous studies have successfully illustrated multiple instructional interventions that could stimulate the AOD design. Nevertheless, limitations existed. First, a few studies examined the holistic design process of an AOD. Many of their studies on AOD design were fragmented, focusing only on one segment (e.g., facilitation). AODs were rarely treated as integral cases wherein a theoretical framework could direct the design from the beginning to the end. Second, the design processes of AODs were never inspired by the cooperative learning theory, even if AOD was expected to support cooperative learning from its origin. Without support from cooperative learning theory, the observation of the AOD design cases lacked a solid, comprehensive, and conceptual perspective.

## 3. Methodology

*3.1. Study Design*

This study is an instrumental case study with both qualitative and quantitative approaches involved. As Creswell (2002) [32] posits, the "case" can be a single individual, event, or activity. When a single case is collected to illuminate a particular issue, the case study is defined as an "instrumental case study" [32]. In this study, one AOD was collected to illuminate if the design of the AOD was innately consistent with the conditions of cooperative learning and if students successfully delivered the expected learning outcomes in the AOD.

*3.2. Study Context*

The context of the AOD was a 15-week graduate-level online course exploring the use of instructional technologies in educational settings, which was taught by a faculty member of the instructional design field in a northeast university in the United States. During this course, students learned various digital technologies and how to use them in practice. The course was offered in a broadly used Learning Management System (LMS) with a built-in AOD board. To protect students' privacy, the faculty did not collect students' demographic information for further identification purposes. A total of 26 graduate students registered for the course.

The instructional goal of the AOD was to prompt students' higher levels of cognitive learning through their interactions on the various topics that surround the use of instructional technologies (e.g., "online resources for educators", "web 2.0"). The students were required to participate in discussions on 14 topics in weekly discussion forums. For the convenience of managing the AODs, students were assigned into two groups (i.e., the blue

team and the red team), and each group included 13 students. While all 13 students of the blue team finished the course, three of the red team dropped the course and did not attend the AODs. Thus, the size of the blue team was 13, and the size of the red team was 10. The members of each team were unchanged throughout the whole semester. Each week, students discussed the same topic in their assigned teams. Students were assigned three or more pre-readings (learning resources) that provided information on the content to be discussed. The AODs were highly student-facilitated, so the instructor and TA were barely involved in the discussion forum.

Figure 2 presents how each AOD was implemented per week. Two students in each team were selected to be moderators. The moderator role would be rotated weekly inside the team to make sure that each student had an equal chance of achieving the experience of facilitating the discussion. Each student was notified of their assigned two weeks for playing the moderator at the beginning of the semester. The moderators had to review the learning resources and prepare facilitated questions as two discussion prompts in two postings when each week started. There was no guidance or rubric introducing which type of facilitating questions should be posed because the instructor intended to motivate students to draft the questions by using their own mindset instead of any existing frameworks. The moderators were also allowed to use their own way to facilitate the discussions since no instruction was provided to pre-define the facilitating strategies. Students who were not moderators (i.e., participants) were required to respond to at least one post from a moderator (i.e., facilitated question) and one from a classmate during the week.

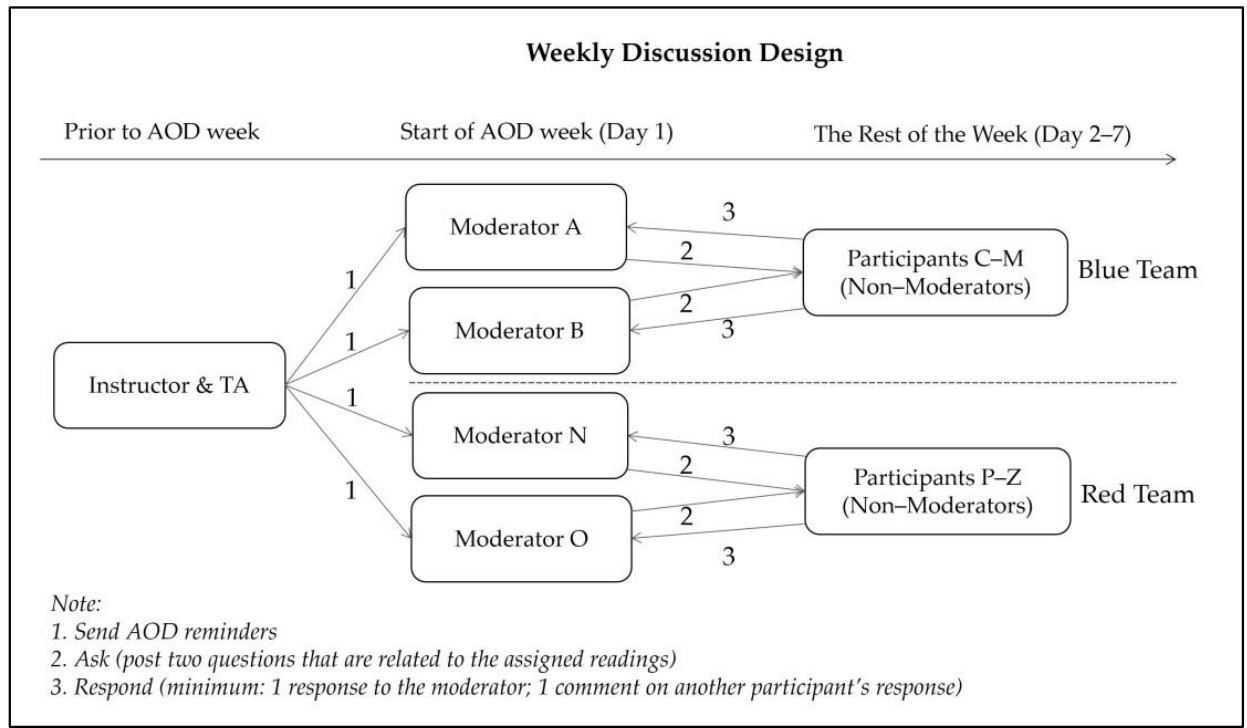

**Figure 2.** Weekly AOD Design.

*3.3. Ethical Statement*

This study received approval from the Institutional Review Board (IRB) of the university to confirm the safety of the participants. All the data were anonymous and untraceable to protect participants' privacy.

*3.4. Data Collection*

To explore the case in-depth, researchers collected multiple data formats, including discussion layouts, course documents (e.g., syllabus, activity instructions), and discussion transcripts. To confirm that students understood the requirements of the AOD, researchers collected their AOD discussion transcripts in the last week of the semester. The discussion topic was eLearning. In the discussion forum, 116 postings were generated by 23 students. In each team, two students played the role of moderators, and each of them generated two postings as initial facilitating questions (i.e., eight postings in the AOD). The 13 participants of the blue team (i.e., participants C–M) generated 40 postings responding to either the facilitating questions or peers' responses, and the 10 participants of the red team (i.e., participants P–Z) generated 68 postings.

*3.5. Data Analyses*

Three data analysis approaches were conducted to answer the two research questions correspondingly. Document analysis was employed to detect how the AOD was designed in the online course and whether the AOD met the conditions of cooperative learning. Document analysis was selected because it is a systemic approach for reviewing, interpreting, and evaluating different formats of materials [33]. It is highly adaptable to studies that focus on understanding conditions in a specific context [33]. The document analysis in the study followed the steps based on Bowen's approach (2009) [33]: (1) creating a cooperative learning checklist as an observation tool, (2) skimming the content of the documents for a superficial examination, (3) reading the content for a thorough examination, (4) filling out the observation checklist, and (5) interpreting the content of the checklist. The materials for the document analysis included the discussion layouts and course documents on the LMS.

A descriptive content analysis was conducted to explore students' cognitive learning patterns in AODs. Descriptive content analysis is an efficient quantitative method for transforming' texts in the AODs into countable constructs by coding the discussion transcripts [34]. The unit of analysis can be varied depending on the instructional context. In this study, every single posting was selected as the unit of analysis, as instructors usually graded students' performance in an AOD by counting and evaluating their postings [35]. A coding framework based on Bloom's taxonomy was developed and employed to count the levels of cognitive learning that occurred in AODs [36,37]. This framework was used to code either participants' descriptive responses or moderators' facilitating questions because both two linguistic formats could indicate students' levels of cognitive learning in the discussion forum [38]. To strengthen the reliability of the coding framework, two researchers independently coded one of the AODs, discussed inconsistencies in coding, and finalized the coding framework. The interrater reliability of the coding framework was 0.8. The coding framework is displayed in Table 1.

At last, chronological visuals were developed to help the researchers observe interactive patterns in the AODs. Inspired by Hmelo-silver et al. (2011) [39], chronological visuals helped them inspect how interactions flowed as time passed. With customizable graphics, chronological visuals provided more dynamic interactive patterns than other approaches. Moreover, since in AODs, cognitive learning happened outside of interactions, this study involved the levels of cognitive learning as new graphic elements to the chronological visuals so that the learning outcomes in the AOD could be better observed and interpreted.

**Table 1.** Coding framework of cognitive learning.

| Code | Indicator | Example |
| --- | --- | --- |
| 0 Level | Simply agree or disagree without further explanations; posting threads that have no relation to topic | Narration: "I especially liked your comment on blended learning definition." Inquiry: "How do you feel about the project so far?" |
| 1 Remembering | Cite sentences (definitions, fact, etc.) related to assigned resources or others' responses; retrieve information from resources outside the course; pose an opinion without any evidence to defend it | Narration: "I agree with the statement in the conclusion where they state . . . " Inquiry: "What are three characteristics of synchronous instruction?" |
| 2 Understanding | Paraphrase authors' thoughts related to the topic; classify different concepts or tools related to the topic; summarize the content of assigned readings or other students' responses; use an example to explain a concept | Narration: "Yamagata-Lynch's example of how she organized her class with asynchronous communication throughout the semester . . . " Inquiry: "How do you understand O'Hear's statement about podcasting?" |
| 3 Applying | Describe how the content of the assigned resources or other students' responses can contribute to real life practices | Narration: "I would offer for my students a certain time of the week . . . " Inquiry: "If you were an instructor of an online math course, how would you use asynchronous and synchronous instruction in the course?" |
| 4 Analyzing | Compare concepts in a scenario; list the strengths or/and weaknesses of content in scenario | Narration: "The only way I can further define it is by comparing it to another learning model, a flipped classroom . . . " Inquiry: "What are the strengths and challenges of using asynchronous strategies in this scenario?" |
| 5 Evaluating | Evaluate a concept or fact by providing strong reasons or evidence | Narration: "On the high school level, I believe that synchronous online classroom settings are more efficient because . . . (giving reasons) . . . " Inquiry: "Which format do you think is better in this scenario and why?" |
| 6 Creating | Give a new idea or plan with strong reasons | Narration: "One way that professors can increase presence and engagement in their class is . . . (giving reasons) . . . " Inquiry: "Devise your own description of distance education using the ideas of synchronous and asynchronous techniques . . . explain your rationale." |

Note: The coding framework was built based on Bloom's taxonomy posited by Bloom et al. (1959) [37] and developed by Anderson et al. (2001) [36].

## 4. Results

### 4.1. Fitness to Cooperative Learning

Through a document analysis of the course syllabus, discussion layout, time schedule, and activity instructions on the LMS system, the researchers found that the design of the AOD only met a few conditions of cooperative learning. Table 2 shows the completed checklist of the AOD's fitness for cooperative learning.

**Table 2.** Checklist of the AOD's fitness for cooperative learning.

| Condition of Cooperative Learning | Content of the Condition | Whether the AOD Design Met the Condition | Evidence Shown in the Course Documents |
|---|---|---|---|
| Expected Learning Outcomes | Interactions; Higher Levels of Cognitive Learning | Positive | Achieving higher levels of cognitive learning within interactions were implied to be the expected learning outcomes in the syllabus. |
| Essential Requirements | Small Group | Negative | Each team included at least 10 students, which did not meet the minimum requirement of the group size (i.e., seven) for a small group. |
| | Equivalent Efforts | Positive | Students had equal chances of playing moderators and had the same workload as participants. |
| | Common Goals | Negative | Students did not need to create specific common products through AOD activities. |
| Cooperative Learning Principles | Positive Interdependence | Neutral | There were clear group boundaries in the discussion forum layout. However, each student's performance was individually graded and barely relied on peers' role-play and content. |
| | Individual Accountability | Neutral | All the postings were visible to the students in the discussion forum. However, there was no clear interface helping students observe each other's contribution to the AOD. |
| | Promotive Interaction | Neutral | Students were heterogeneously grouped. Yet, there were no instructions on how to promote interactions among students. |
| | Social Skills | Negative | No training or materials of social skills were provided for students. |
| | Group Processing | Negative | Students were not required to reflect on their group work during the AOD. |
| Practice | Structure | Negative | The AOD was designed as a one-step activity. There were no clear instructions for sequential behaviors from students. |

In the course syllabus, the learning objective in the AOD was indicated as achieving higher levels of cognitive learning, including "analyzing," "evaluating," and "creating," according to Bloom's taxonomy [36,37]. Each student's workload and role-play chance in the AOD were equal whenever the student played a participant or moderator. Thus, students

in the AOD were expected to obtain cooperative learning outcomes with equivalent efforts in cooperation.

Yet, the design of the AOD did not fit the rest of the cooperative learning requirements. Although the students were assigned to two discussion teams (i.e., the blue and red teams), the size of a single discussion team did not meet even the most flexible group size requirement (i.e., less than seven) of the cooperative learning theory [8]. Moreover, the students did not share common goals in the AODs. They were required to participate in the AODs by posting responses, but there was no evidence proving they had to create one or more common visible products through the AODs. Postings could be regarded as visual products reluctantly, but a posting could be finished by an individual without cooperation. Thus, creating postings could not be a common goal shared by all the students from the same group.

The positive interdependence principle was not confirmed in this AOD design. The interdependence was only substantially positive in the boundary dimension. With the technological support, the discussion forum in the LMS system had two separate discussion spaces for the two teams so that students could easily define who was interdependent with whom. In this case, however, the instructor graded each student's performance separately, thus preventing reward interdependence. As for the means interdependence, participants' role performance depended on the moderators' role performance, as the participants were required to answer the initial facilitating questions constituted by the moderators. However, the moderators performance did not seem to depend on the participants' performance because there was no requirement for moderators' to facilitate behaviors. Further, in the AOD, students were not required to share each other's content to construct their postings, and they could construct their postings individually without reviewing the contents of others.

Even though each student could review all the peers' postings in the AOD, the AOD design did not provide students with a straightforward and visualized approach to easily recognize each other' contribution to an AOD, especially in a forum with more than ten students involved. Therefore, students could not realize individual accountability in collaboration with their peers because there was no design feature to facilitate this. In order to trigger promotive interactions, the instructor carefully formed the groups by making the backgrounds of members diverse in the same group. However, the instructions of the AOD did not deliver the methods of promoting interactions, such as exchanging materials, encouraging each other, providing feedback, and giving sample postings [13]. Hence, it was hard to ensure that students would promote interactions.

Moreover, no content knowledge or intended practices of social skills were available for students in the AODs. It was difficult to confirm that students were well-prepared with some social skills, such as decision-making or conflict management [19]. Further, the design of the AODs did not lead students to reflect on their cooperation in the AODs. There was no space left for students to discuss the cooperation issues occurring in the AODs. It meant that students could rarely spare time to examine their cooperation performance and explore how their cooperation in the AODs could be conducted more effectively.

As the design of the AOD was not strongly inspired by cooperative learning, the AOD was finally constructed without specific structures. This meant that the AOD activity was not organized as several sequential steps for the convenience of meeting multiple conditions of cooperative learning. Students' participation and cooperation in the AOD were expected to be completed spontaneously without guidance. Although the learning objective was clearly stated, the programming of having access to the learning objective was oversimplified.

### 4.2. Patterns of Cognitive Learning

Lower-level-dominant. Overall, the number of postings that demonstrated higher levels of cognitive learning was far below the total number of postings. Only 31 postings out of 116 demonstrated higher levels of cognitive learning. Figure 3 depicts the distribution of

cognitive learning levels. In the comparison between the performance of the blue team and the performance of the red team, the latter had slightly more postings with higher levels of learning (17 vs. 14), but its percentage of postings with higher levels of cognitive learning was lower than that of the blue team (24% vs. 32%). Therefore, both teams performed poorly when demonstrating higher levels of cognitive learning. As the design of the AOD included moderators and participants, the researchers decided to further detect if the two roles performed differently in delivering postings with higher levels of learning. It proved that the four moderators from the two teams created five initial facilitating questions with higher learning levels, which took up almost 63% of the initial facilitating questions (i.e., five out of eight facilitating questions). It illustrated that they achieved higher levels of cognitive learning when they drafted facilitating questions.

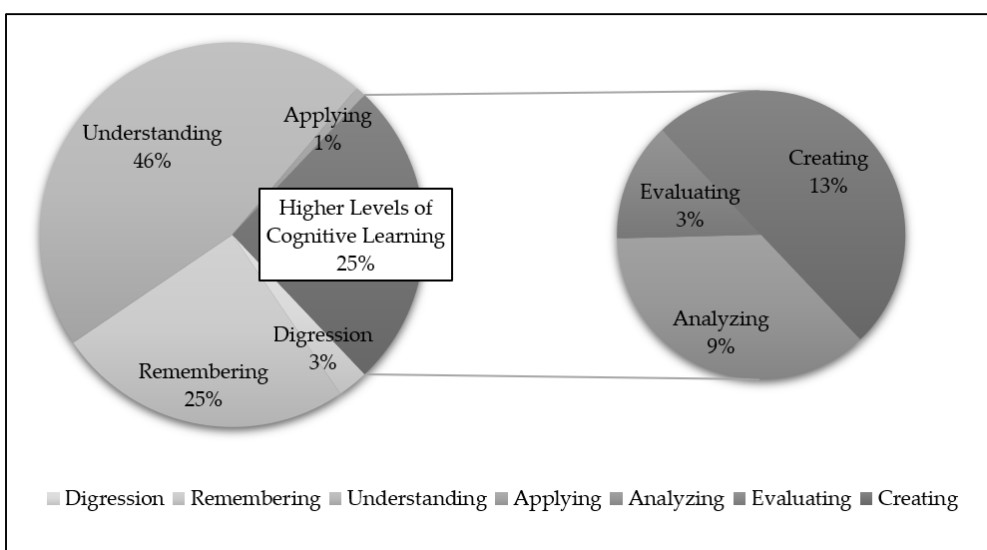

**Figure 3.** Distribution of cognitive learning levels in the AOD.

However, moderators performed badly when delivering postings with higher levels of cognitive learning when they left moderating postings in the rest of the AOD. Only 4% of their moderating postings (i.e., one out of 23 moderating postings) stayed at higher levels. Comparatively, participants performed much better in terms of their responsibility of participating in the AOD. A total of 35% (*n* = 85) of their postings reached higher cognitive learning levels. In summary, while moderators created a decent number of initial facilitating questions triggering higher levels of learning, they did not continue to promote the discussion by contributing postings with higher levels. Participants were more engaged in the discussion but still displayed a limited number of postings with higher levels of learning within the scope of their main responsibility.

Contribution-skewed. Apart from the low proportion of postings with higher levels of cognitive learning produced in the AOD, the contribution of postings with higher levels from the participants was also highly skewed. As Figure 4 displays, only five participants contributed three or more postings with higher levels of learning in the AOD. When perceived from a bigger picture, they contributed more than 50% of the postings with higher cognitive learning levels while only representing about 20% of the sampled population. Comparatively, eight participants did not exhibit any higher levels of cognitive learning in their postings. Therefore, the data revealed that some students did not achieve higher levels of learning in the AOD. The design of the AOD did not maximize each individual's cognitive learning as cooperative learning requires.

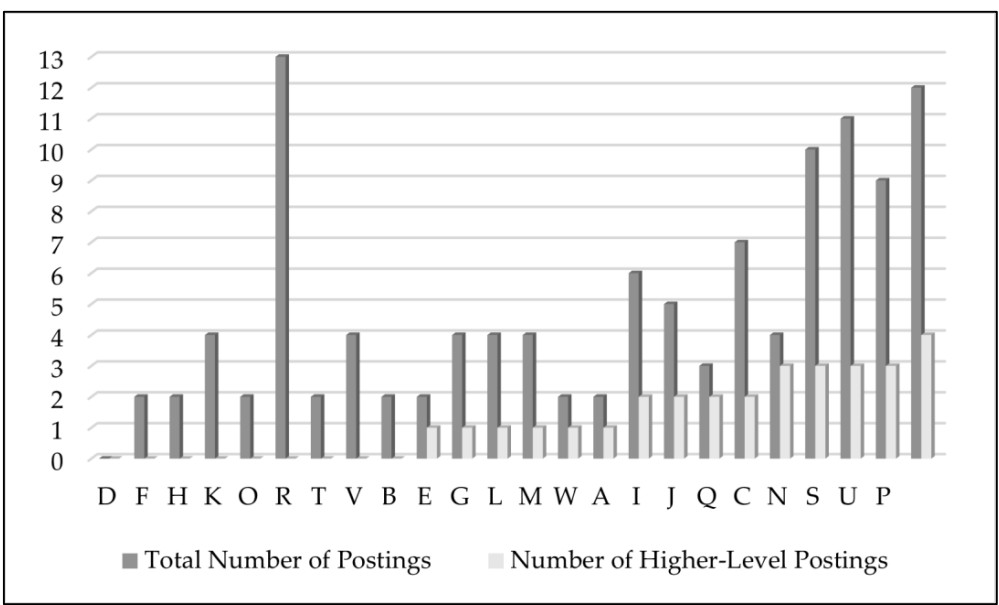

**Figure 4.** Higher-learning-level postings (*n* = 30) created by each student.

Distribution-imbalanced. In a cooperative learning activity, students are expected to achieve higher levels of cognitive learning, but it does not mean the lower levels of cognitive learning are not worth researchers' attention. Lower cognitive learning levels are also indispensable processes in retaining and constructing knowledge [36,37]. Therefore, this study inspected all the levels of cognitive learning that occurred in the AOD. 25% of the postings stayed at the "remembering", and 46% of the postings stayed at the "understanding" level. It indicated that in the AOD, students could excel at identifying, retrieving, and interpreting information from multiple sources. Yet only 1% percent of postings stayed at the "applying" level. This suggested that students barely transferred the information to real-life examples or inferred an issue from a practical perspective. The shortage of the "applying" level exposed that students did not actively demonstrate their problem-solving cognitive skills in the AOD. Among the three higher levels of cognitive learning, the "creating" level took a larger proportion (i.e., 13%) than the rest of the two together (i.e., 12%). The dominance of the "creating" level over the other two higher levels manifested that, in the AOD, it is more natural for students to draft new ideas with evidence or reasons. Meanwhile, students were not inclined to make a comparison or evaluate an object without navigation embedded in the design of the AOD.

*4.3. Patterns of Interaction*

Figure 5 shows the chronological visuals analyzing students' patterns of interaction in the AOD. When the researchers of this study examined the interactions that occurred in the chronological visuals, they counted the number of different interaction indicators suggested by [40]. Andrew coined how analyzing an AOD should depend on five interaction indicators: (1) the number of single postings, (2) the percentage of students participating in the AOD beyond the minimum required number of postings, (3) the number of developed threads, (4) the number of participant–participant conversations, and (5) the number of participant-moderator/instructor conversations [38,40]. According to the chronological visuals, 116 postings were created by moderators and participants, and 65% of students actively created postings in the AOD beyond the minimum required number of postings. The two indicators suggest that students' active participation provided a good foundation for meaningful interactions in the AOD. When the other three indicators with the chronological visuals were reviewed, interactive patterns appeared.

The Legend Explaining the Meaning of Each Symbol

| Moderator's posting | A | Understanding | □ |
|---|---|---|---|
| Participant's posting | E | Applying | ◇ |
| E Replied to J | J / E | Analyzing | ▲ |
| Digression | ⊗ | Evaluating | ■ |
| Remembering | △ | Creating | ◆ |

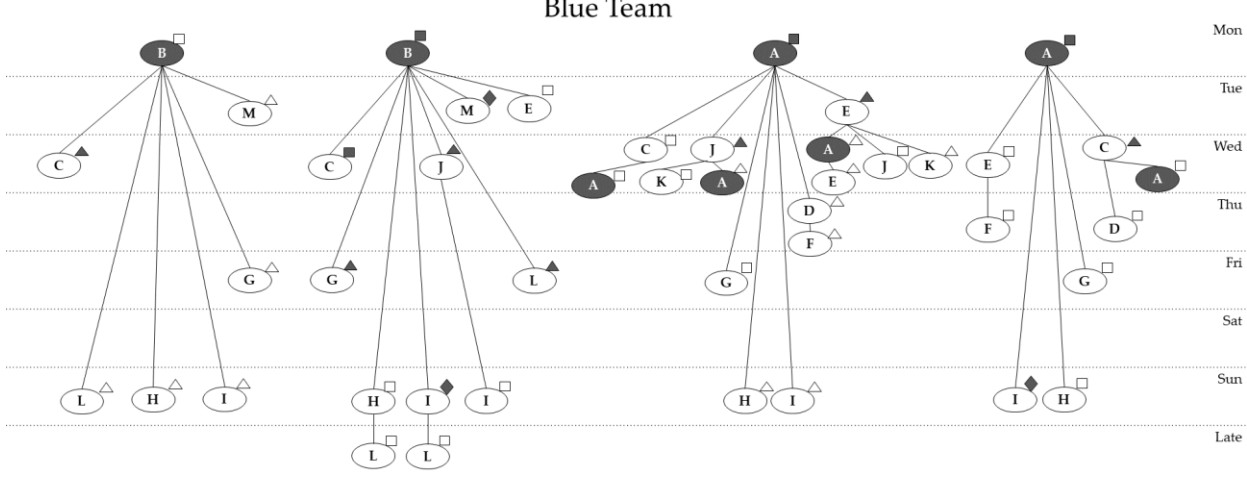

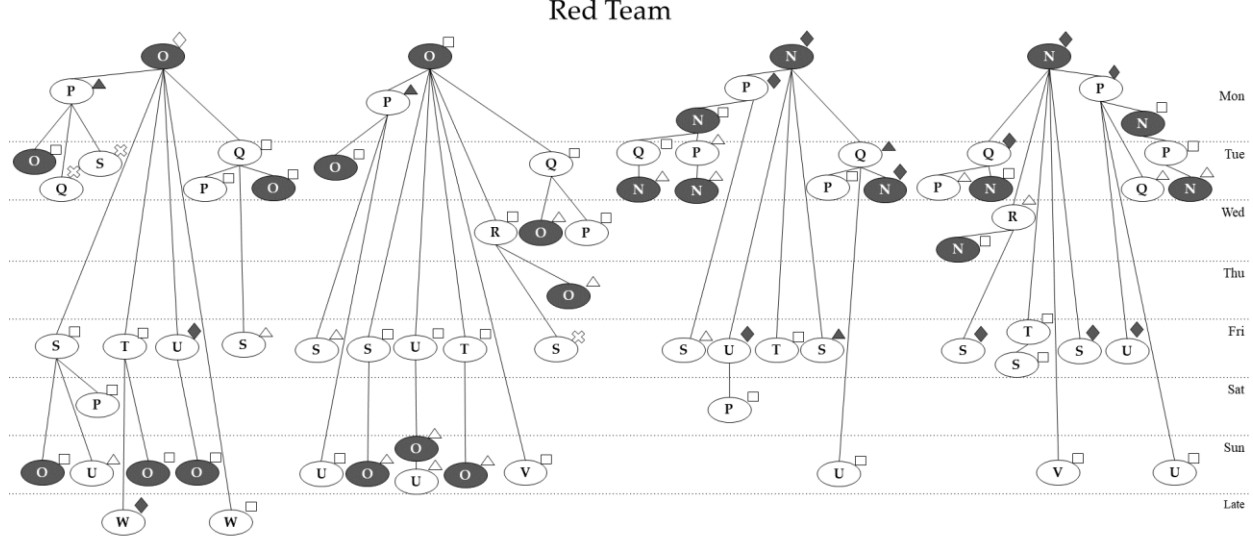

**Figure 5.** Chronological visuals.

Moderator-centered. In the chronological visuals, one circle with a letter in the center represents a single posting created by a moderator or participant. The graphic of a line connecting two circles represents a single conversation. Thus, the chronological visuals exhibited 107 one-on-one conversations in total. Only 29 conversations happened between the two participants. Seventy-eight conversations happened between a participant and a moderator, making up 73% of the total conversations. This demonstrated that in the AODs, participants tended to interact more with the moderator than with other partici­pants. Moreover, among these participant-moderator conversations, 64% of them were conversations in which the participants were responding to the moderators' facilitating questions as the AOD activity required. The chronological visuals revealed that only a few participants would reply to the moderators' postings in the middle of the AOD. Therefore, it could be concluded that participants mostly interacted with the moderators, instead

of with other non-moderator peers, through question-answer sessions. They seemed to treat the AOD as an assignment to be submitted to the moderators rather than activities with which to interact with peers. This finding was inconsistent with the literature, which proposed that only in an AOD with an instructor-led facilitation strategy applied would students participate in the AOD by submitting their postings as assignments [4,41]. This pattern showed another possibility about AOD. Students could still regard their postings as assignments even if the AOD was designed with a student-led facilitation strategy because the "moderator" could be designed as a simulation role of the instructor bringing similar effects to students' performance in the AOD.

Underdeveloped. In chronological visuals, a student–student interaction can be a single one-on-one conversation from the beginning to the end. Or it can be a developed thread with multiple conversations involved with or without a meaningful sequence. In total, 50 interactions were found in the AOD. Almost half of the interactions (46%) were underdeveloped as single conversations. For example, in the first discussion thread moderated by students B, C, L, H, I, G, and M all participated in the blue team in the form of one-round conversations. These single conversations did not evolve into threads with more follow-up conversations. Observing the developed threads, the researchers found that 74% of the threads benefited from the moderator's facilitation. Furthermore, only 20% of these developed threads included participants' iterative engagement. It meant that generally, participants would not continue to talk to the same person in a thread and develop an in-depth one-on-one conversation. Instead, they quit the conversation thoroughly after leaving one response without caring where the conversation would be directed to. This finding matched the previous observation that students mostly regarded the AOD as a one-time writing assignment rather than an ongoing instructional activity. As a large amount of one-round conversations and a limited number of participants' iterative engagement existed, it was hard to claim that students' interactions were well-developed in the AOD.

Intermittent. Chronological visuals demonstrated students' interactions and their cognitive learning in a time frame. Figure 6 shows how the number of postings and the percentage of postings with higher levels of learning in each team of the AOD changed over time. Regarding interactions, the two teams performed similarly, with minor discrepancies in the timeline. For both teams, the peak of interaction occurred in the early stage of the AOD (i.e., Monday and Wednesday), as the red team created 21 postings and the blue team created 15 postings on Wednesday. In the middle stage of the AOD (i.e., Thursday), students' interaction in both teams dropped dramatically. On Friday, the interaction rebounded slightly in the blue team and largely in the red team. Saturday was an expectable valley of interaction as the AOD proceeded to the weekends. However, both teams posted a decent number of postings on the last day (i.e., 13 postings from the red team and 10 postings from the blue team), which could be suspected of cramming behavior. The interaction changed over the week and implied that even though the AOD provided students with a very flexible schedule, most interactions happened intensively in a few narrow time ranges. Zooming in on the change in interaction quality, the researchers further found that the tendencies of the two teams' cognitive development almost overlapped and were consistent with the tendencies of their interactive development. For instance, on Monday, for both groups, the number of postings with higher levels of learning took up more than 50 percent of the total number of postings. Another two peaks occurred in the middle and the end of the week (i.e., Friday and Sunday/Late). It proved that both interaction and cognitive development were not continuous but intermittent in the AOD.

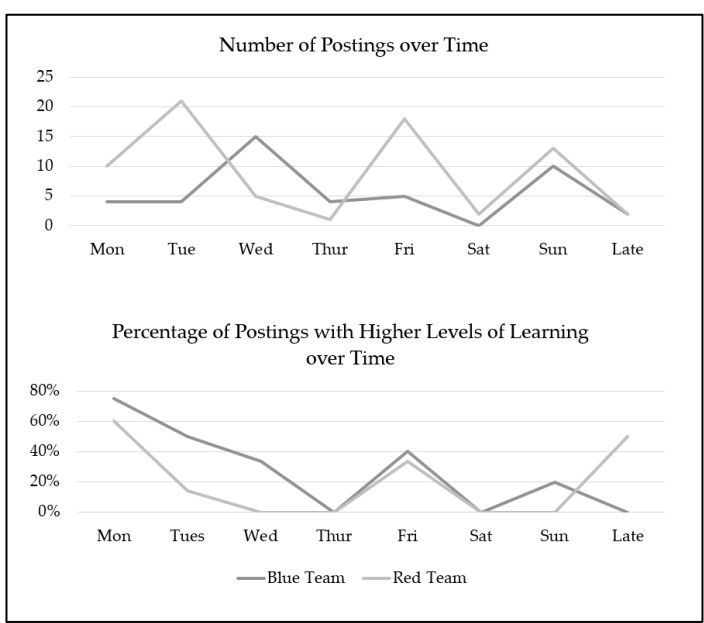

**Figure 6.** AOD performance change over time.

### 5. Conclusions

This study investigated an AOD design case from the perspective of cooperative learning. The results suggested that AOD had great potential to support cooperative learning in distance education due to its gifted merits. However, it could not help students achieve the expected learning outcomes when cooperative learning conditions were not translated into the instructional design process. In the AOD design case, students demonstrated a lower level of cognitive learning and did not contribute equal effort to the cooperative work. Some important levels of cognitive learning were missing in constructing meaningful interactions. In addition, students would interact with each other in the AOD, but their interactions were primarily moderator-centered and lacked conversations between participants. Most of the conversations in the AOD did not expand further. Students rarely extended a one-on-one conversation and often participated in the AOD intermittently during the week. These interaction patterns indicated that students' participation and cooperation in the AOD were passive, task-oriented, and less motivated.

This study could also enlighten the AOD design by analyzing its fitness for cooperative learning. According to the results, a few conditions of cooperative learning could be automatically met in the AOD. For example, supported by computer-mediated asynchronous communication tools, instructors could easily build up group boundaries though the visualized layout of the discussion forum so that students' positive interdependence could be enhanced; students' contributions to the discussion were visible to all, which could reinforce students' feeling of individual accountability. The finding was matched with the literature stating that AOD had great potential to be a cooperative learning activity due to its inborn advantages [1,2,4,42]. The results showed that most of the conditions needed to be operationalized through the instructional designer's active investment in the design process to actualize AOD's potential. The instructional designer in this AOD established several valid design features meeting some cooperative learning conditions: expected learning outcomes were introduced in the syllabus; workload and tasks were equal; students were heterogeneously grouped [8,13,17]. Nevertheless, more expected design features for cooperative learning were missing in the case. In the future of AOD design, the instructors or designers could consider adding more design features in alignment with the conditions of cooperative learning. Table 3 demonstrates some potential suggestions supporting AOD as a cooperative learning activity. Some of them can be highlighted. The size of each cooperative group should be smaller than seven [2,8]. The instructor may integrate some design features, such as a feedback system (e.g., performance reports), into

AODs to confirm that students can realize their individual accountability [6,11]. Prior to AOD, students may need to receive training to improve their social competency so that they can interact with each other empathetically rather than only regard postings as missions [17,19]. In addition, an AOD could be more structural, with multiple meaningful steps involved [15,25].

**Table 3.** Potential design features supporting cooperative learning in AODs.

| Condition of Cooperative Learning | Content of the Condition | AOD Design Features Corresponding to Cooperative Learning |
| --- | --- | --- |
| Expected Learning Outcomes | Higher Levels of Cognitive Learning within Interactions | The expected learning outcomes should be clearly delivered in the activity instructions or grading rubrics [6,28] |
| Essential Requirements | Small Group | The number of each group should be smaller than seven [2,8]. |
| | Equivalent Efforts | The workload and role-play chance of each group member should be equal [2,14]. |
| | Common Goals | Students are required to create common products in the end of the AOD, such as a discussion summary or a problem solution [17]. |
| Cooperative Learning Principles | Positive Interdependence | Students' performance can be graded as a group. The roles' responsibilities (e.g., moderator, participant) should depend on each other [6,11]. |
| | Individual Accountability | The instructors can create and send all the group members' discussion performance reports to the group [19]. |
| | Promotive Interaction | Students can be heterogeneously grouped. Students can receive guidance about how to construct meaningful interactions [20,21]. |
| | Social Skills | Resources about social skills can be provided. Students can be trained for the use of social skills [17,19]. |
| | Group Processing | Students need to have a chance of sharing their thoughts on the group work in the AOD [13,15]. |
| Practice | Structure | Multiple structures can be applied, such as Paraphrase Passport, Think-Pair-Share, and Jigsaw [15,25]. |

## 6. Limitation and Future Research

This case study only explored a single AOD and was conducted in a graduate-level online course. Therefore, the generalizability of the results is limited, and more factors, such as students' educational level, should be considered in future studies. However, it contributes an insightful perspective on observing AODs, especially when associated with the cooperative learning theory. Interpreting an AOD under the cooperative learning scheme can inspire other researchers to diagnose the issues about AODs and systematically speculate about possible changes in the current AOD design. In future studies, the approach of observing the AOD applied in this study could be developed into a concise and comprehensive evaluation tool (e.g., rubric) to help instructors evaluate if an AOD meets the conditions of cooperative learning.

More importantly, this study can also inform the development of cooperative learning in the new digital age. Despite having been practiced in face-to-face settings for a couple of years, cooperative learning needed design cases with empirical evidence that supports its value in distance education. More studies are needed to confirm if an AOD can meet all the cooperative learning conditions and successfully help students achieve higher levels

of cognitive learning and interact with each other frequently. Therefore, the conditions of cooperative learning should be involved and considered at an early stage of designing an AOD. In the future, this study will aim to design AODs meeting all the cooperative learning conditions and examine students' performance in a real cooperative learning setting.

**Author Contributions:** Conceptualization, T.Y.; methodology, T.Y.; software, T.Y.; validation, T.Y.; formal analysis, T.Y.; investigation, T.Y.; resources, T.Y.; data curation, T.Y.; writing—original draft preparation, T.Y.; writing—review and editing, Z.N. and T.Y.; visualization, T.Y.; supervision, T.Y.; project administration, Z.N. and T.Y.; Funding acquisition, T.Y. All authors have read and agreed to the published version of the manuscript.

**Funding:** This research received no external funding.

**Institutional Review Board Statement:** The study was conducted in accordance with the Declaration of Helsinki, and approved by the Institutional Review Board of Syracuse University. The protocol code is 17-027 and approval date is 25 January 2017.

**Informed Consent Statement:** The student consent was waived because the Institutional Review Board approved the exempt protocol indicating the study used untraceable and anonymous data which already existed before the study was conducted.

**Data Availability Statement:** Data sharing is not applicable to this article.

**Conflicts of Interest:** The authors declare no conflict of interest.

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
