# Peer review of "Investigating the Design of an Asynchronous Online Discussion (AOD) in Distance Education: A Cooperative Learning Perspective"

_education, doi:10.3390/educsci13040412_

Round 1

Reviewer 1 Report

The authors present a descriptive, observational case study linking Asynchronous Online Discussion in Distance Education to Cooperative Learning Theory for the first time. They establish the gap in the literature and propose logical research questions.

The article is timely given the evolving balance of blended: face to face provision now underway across the sector post-pandemic. The findings will be helpful and applicable to many practitioners globally.

The abstract is a clear, well written summary of the work.

Suggested revisions:

The introduction requires re-structuring to ensure that the Theoretical Framework is introduced earlier in a more substantial way in order to justify the research questions that follow (e.g. citations in line 29 and a brief description of the theory prior to research questions).

Suggest that Fig1 is moved to later in the Theoretical Framework section. Where it is, with no prior citations/ explanation implies that it is novel and not merely a visual summary of existing theory. It also requires a figure legend and some references to it in the text.

The description of the theoretical framework is helpful, however, it may be possible to do this more concisely.

Parts of The Expected Learning Outcome section actually refer to the  Essential Requirements. This section also needs minor development to explain the following statement (line 58): ..students are expected to achieve higher levels of cognitive learning through interactions.. (compared to what?).

The methodology is largely sound and well justified. However, the study concerns human participants (the students), and although an observational case study in design (there is no intervention group), in the reviewer's institution, this would require ethical approval. Please include an ethics statement to explain what ethical approval is in place for the study, or to justify the lack of one, describing how participant interests were safeguarded.

The results are comprehensive and well presented.

The conclusions are largely descriptive in nature, and one criticism would be a lack of criticality towards the underpinning cooperative learning theory and its appropriateness for the distance setting. There has been no attempt to find links to existing literature relating to cooperative learning or the distance learning context, indeed there are no citations in the main text.  I appreciate a lot of this information is summarised in table 3, but this would benefit from some discussion.

The limitations are well framed, however the proposed future research could be further reaching and test the assumption that AOD that completely fulfills the cooperative design brief will support interaction and cognitive learning in the distance context.

Reviewer 2 Report

Although there are countless works on collaborative learning, I found the approach taken by the authors in this study very interesting, thorough, and comprehensive. In addition, I found the article very easy and enjoyable to read.

After minor revisions, I recommend it for publication in the “Education Sciences” journal. I hope the authors take these corrections in good faith, as I intend their work to be published with the best possible quality. From now on, I will use this format: P=page, L=line, example: P2 L33.

·      Did the paper's authors evaluate this AOD case externally, or did any of them implement it? There are certain variables that, although easily controllable, were not controlled, such as the behavior of the moderators (e.g., the development of trigger questions), the group size, or establish common goals.

·      Avoid repeating both the expressions “previous studies” and “previous researchers.”

·      Be careful using repeated words within the same paragraph and throughout the document. For example, avoid repeating “case,” “case studies,” “researcher,” “week,” and “students.”

·      Check the font in all tables of the document.

·      P1 L17. The name “Asynchronous Online Discussion” and its acronym are repeated on the same line.

·      P1 L39. The first research question could be reformulated to have more in-depth answers rather than a yes-no answer. For example: What features make the AOD case design consistent with cooperative learning conditions?

·      P2 L61. Review the use of colons.

·      P3 L105-114. In this paragraph, there are many repeated words.

·      P3 L111-114. I suppose the references (Johnson & Johnson, 2014) and (Johnson & Johnson, 2014) are the same. Unify.

·      P3 L114. The expression “et al” should be written as “et al.”.

·      P3 L221. Could you give a better context of the case study? University, faculty, a subject where the study was implemented, how many participants were involved, sociodemographic data, etc.

·      P6 L236. How many people were on each team? Did the teams change every week, or were they the same throughout the semester?

·      P6 L242. Were there any rules for choosing moderators? For example, each week, they must rotate the 2 selected by the team. Table 2 says that students had equal chances of playing moderators. Why is that?

·      P6 L242. Were there any rules for the moderators’ behavior? Did the instructor or TA assess whether the questions asked by the moderator could contribute to higher levels of cognitive learning?

·      P6 L259. If the recommendations made in the bibliography speak of small teams of between 2 and 7 people (P2 L68), why did you make only two teams of 10 and 13 students each?

·      Table 1. 5. Evaluating. Narration: The ellipses should be expressed as “…”. Also, I would add “because” in the sentence: “On the high school level, I believe that synchronous online classroom settings are more efficient because… (giving reasons) …”

·      Table 1. 5. Evaluating. Inquiry: Write “do” instead of “to” in: “Which format do you think is better in this scenario and why?”

·      Table 1. 6. Creating. Rewrite the sentence avoiding repetitions.

·      P8 295. Reference (Hmelo-silver et al., 2011) should not be twice.

·      Table 2. Equivalent Efforts. “Students have equal…” Past tense should be used.

·      P10 L375-379. I'm not sure what this whole phrase means. I'll go in parts.

o   a. “It proved that the four moderators from the two teams created five initial facilitating questions with higher learning levels” -> It means that every week, the elected moderators created questions with a higher level of learning; is this correct?

o   b. "which took up 63% of the initial facilitating questions” and “Only 4% of their moderating postings (n=23) stayed at higher levels.” -> How do you calculate these values?

o   c. “However, moderators performed badly in delivering postings with higher levels of cognitive learning when they left moderating postings in the rest of the AOD.” It means that after the moderators ask the questions and the participants respond, the moderators don't do a good job of "moderating" those responses. I have a question: Was there any instruction on moderating the participants' answers?

·      Figure 3. Check the word “cognitive” in the left pie chart.

·      P10 L394. The expression “more than three” should be “three or more.”

·      Figure 4. The title could be “Higher-learning-level postings (N=30) created by each student” to reinforce that there were 30 postings, not 30 students. The sentence should not be capitalized, and check the font size.

·      Figure 4. The total number of posts made by each student could be added in another color to show the ratio of Higher-Learning-Level Postings over the total number of postings.

·      Figure 5. It should say “Blue team” and “Red Team”. Also, the word "Legend" above the table has no explanation.

·      Figure 5. I understood the moderators' initial post would be a trigger and facilitator question. Why do authors also classify questions within Bloom's taxonomy?

·      P13 L467. The sentence “For example, there were several six student-student interactions…” should be rewritten. By the way, I count just five student-student interactions in that example: B with C, G, L, M, and E.

·      Figure 6. How are the percentages in Figure 6 interpreted? For example, 75% of the posts made by the Blue Team on Monday have been of a higher learning level; is this correct? A brief explanation of this image could be made in the text.

·      P14 L503. Check the sentence: “…merits, However…”

·      P14 L511. Check the sentence “…the same patterner and…”

·      Table 3. The column “AOD Design Features Corresponding to Cooperative Learning” should be aligned to the left.

·      Limitation and Future Research. Another limitation is that this study was conducted in a graduate-level online course. It would be interesting to see what characteristics are met at other educational levels and make a comparison in this regard.

·      The bibliography must be revised. The phrase “Retrieved from” should not be used. Use of dash line (“−“, not “-“).

·      Bibliography. References must have doi. For example, for the reference “An, H., Shin, S., & Lim, K. (2009)”, the doi is: https://doi.org/10.1016/j.compedu.2009.04.015.

Although the reader may find the article easy and enjoyable, I encourage the authors to revise the manuscript as recommended above for publication. Congratulations on the work done!
